# Organic matter processing by heterotrophic bacterioplankton in a large tropical river: Relating elemental composition and potential carbon mineralization

**Daniel Cuevas-Lara[1], Felipe García-Oliva[2], Salvador Sánchez-Carrillo[3], Javier Alcocer[4]***

**1** Programa de Posgrado en Ciencias del Mar y Limnología, Universidad Nacional Autónoma de México, México City, México, **2** Instituto de Investigaciones en Ecosistemas y Sustentabilidad, Universidad Nacional Autónoma de México, Morelia, Michoacán, México, **3** Departamento de Biogeoquímica y Ecología Microbiana, Museo Nacional de Ciencias Naturales (CSIC), Madrid, España, **4** Grupo de Investigación en Limnología Tropical, FES Iztacala, Universidad Nacional Autónoma de México, Tlalnepantla, Estado de México, México

* jalcocer@unam.mx

**Data Availability Statement:** All relevant data are within the manuscript and its Supporting Information files.

## Abstract

River hydrology shapes the sources, concentration, and stoichiometry of organic matter within drainage basins. However, our understanding of how the microbes process dissolved organic matter (DOM) and recycle nutrients in tropical rivers needs to be improved. This study explores the relationships between elemental DOM composition (carbon/nitrogen/phosphorus: C/N/P), C and N uptake, and C mineralization by autochthonous bacterioplankton in the Usumacinta River, one of the most important fluvial systems in Mexico. Our study investigated changes in the composition and concentration of DOM and evaluated carbon dioxide ($CO_2$)production rates ($C–CO_2$) through laboratory experiments. We compared three sites representing the middle and lower river basins, including their transitional zones, during the rainy and dry seasons. After incubation (120 h at 25°C), the DOM decreased between 25% and 89% of C content. Notably, the initial high proportion of C in DOM in samples from the middle–forested zone and the transition led to elevated $C–CO_2$ rates (>10 mg $l^{-1}$ $day^{-1}$), in contrast to the lower initial C proportion and subsequent $C–CO_2$ rates (<7 mg $l^{-1}$ $day^{-1}$) in the lower river basin. We also found that dissolved organic carbon uptake and $NO_3^-$ and $NH_4^+$ production were higher during the dry season than in the rainy season. The low water flow in the river during the dry season accentuated the differences in elemental composition and microbial processing of DOM among the sites, while the high water flow of the rainy season homogenized these factors. Our findings indicate that microbial metabolism operates with reduced efficiency in C-rich environments like forests, particularly when faced with high C/N and C/P ratios in DOM. This study highlights the influence of the tropical hydrological regime (rainy and dry seasons) and the longitudinal changes in the river basin (middle and lower) topography and land cover on microbial metabolism by constraining DOM characteristics, emphasizing the crucial role of elemental ratios in river DOM processing.

**Funding:** This research was funded by the Fondo Institucional de Fomento Regional para el Desarrollo Científico, Tecnológico y de Innovación (FORDECYT) – Consejo Nacional de Ciencias y Tecnología (CONACYT) in the form of Project "Fortalecimiento de las capacidades científicas y tecnológicas para la gestión territorial sustentable de la Cuenca del Río Usumacinta y su Zona Marina de Influencia (CRUZMI), así como su adaptación ante el cambio climático" [273646 to JA], the Universidad Nacional Autónoma de México (UNAM) – Programa de Apoyo a Proyectos de Investigación e Innovación Tecnológica (PAPIIT) in the form of Project "Flujos de carbono, nutrientes y sedimentos en un sistema lótico tropical" [IN216818 to JA] and Project "Procesos biogeoquímicos que influyen la estequiometría elemental C:N:P en diferentes ecosistemas terrestres mexicanos" [IN207721 to FGO], the Programa de Investigación en Cambio Climático (PINCC), UNAM in the form of Project "Cuerpos acuáticos epicontinentales: papel en la dinámica del carbono y emisiones de gases de efecto invernadero en México" [PINCC 2020 and PINCC 2021 to JA], the Spanish Ministry of Science and Innovation (MCIN/AEI) through the "European Union NextGenerationEU/PRTR" grant [PID2020-116147GB-C21/AEI/10.13039/501100011033 to SSC], and the Ministerio de Ciencia e Innovación, Consejo Superior de Investigaciones Científicas de España within the i-COOP+2020 in the form of Projects "Contribución del neotrópico acuático continental a las emisiones de gases de efecto invernadero" [COOPA20433 to SSC] and "Los grandes ríos del neotrópico y su contribución al ciclo de carbono global" [COOPA20472 to SSC].

**Competing interests:** The authors have declared that no competing interests exist.

## Introduction

Rivers can receive, transform, and outgas significant amounts of terrestrial–derived organic matter. Recent studies indicate that inland waters contribute approximately 0.9 Pg C $yr^{-1}$ to global oceans. This figure is significantly lower than the 2.9 Pg C $yr^{-1}$ originating from terrestrial sources and the 1.9 Pg C $yr^{-1}$ released into the atmosphere [1]. These differences suggest that the decomposition of organic matter by heterotrophic prokaryotes, along with other abiotic processes like photomineralization and terrestrial inputs of $CO_2$ [2], plays a crucial role in the fluvial C evasion, as they mineralize organic matter into $CO_2$ through respiration, especially in the tropics [3, 4]. A large quantity of $CO_2$ flows through tropical inland waters, indicating the importance of microbial activity to C cycling in these regions [5].

The influence of dissolved organic matter (DOM) and inorganic nutrients on C mineralization should change along the rivers. Inputs of nitrogen (N) and phosphorus (P) can either raise respiration rates, reduce biomass, or elicit negligible reactions among planktonic and benthic microbial communities [6–8]. In small upland streams, the assimilation of nutrients from sediment pore spaces amplifies nutrient cycling within the riverbed compared to the water column [9]. Furthermore, the quantity of particulate organic matter (POM) affects the influence of dissolved nutrients on microbial respiration because microbial extracellular enzymes produce monomers with the available nutrients by degrading large biomolecules in the POM [10, 11]. However, recent findings indicate that these metabolic processes predominantly occur within the water column rather than the riverbed in large rivers (fifth order or above). This is attributed to the enhanced surface area for contact between the water and suspended sediments found further downstream [12, 13]. Thus, the influence of stream order on microbial processing of organic matter in the water column needs more research efforts.

The availability of organic matter changes with landscape characteristics along basins and according to precipitation levels [5]. The supply of both labile (e.g., a 5% increase as sucrose) and recalcitrant (e.g., lignin and cellulose) terrestrial DOM can increase respiration and biotic $CO_2$ emissions in upland river reaches bordered by upland forests and soils rich in organic matter [14, 15]. Moreover, primary production in lowland rivers yields short–term biologically reactive organic carbon [16]. These autochthonous inputs are high in the dry season when low precipitation leads to ample water residence time and a low water discharge that prompts primary productivity [17]. In contrast, rainy season precipitation exports terrestrial–derived DOM from the watershed [18]. Precipitation seasonality (rainy *vs.* dry season) can dictate the metabolic rhythms of tropical aquatic ecosystems [19, 20]. Understanding how variation in organic matter supply affects the microbial C mineralization at larger (e.g., cross-regional) scales will help to elucidate global C cycling and how it may respond to climate change [5, 19, 21].

According to ecological stoichiometry, elemental DOM composition (i.e., C/N/P ratios) changes microbial metabolism since an excess or depletion of elements affects the use of ingested C [22, 23]. Microbes must invest energy to adjust these elemental ratios to maintain homeostasis within their biomass, whose growth is usually limited by N, P, or both compared to C according to resource stoichiometry [24–26]. For instance, bacteria prioritize enzyme production over somatic growth to access scarce nutrients [27, 28]. The C/N/P ratios in organic matter significantly affect microbial decomposition and C mineralization rates [29, 30], varying according to the source of organic matter along rivers. Autochthonous inputs, like phytoplankton, exhibit low C/N and C/P ratios, whereas allochthonous inputs, such as wood and leaves, feature high C/N and C/P ratios [31]. Although studies of ecological stoichiometry have shown the effect of nutrient depletion on microbial metabolism, they have not yet addressed the consequences of stoichiometric imbalance due to an elevated proportion of one element over the others [32, 33].

The Usumacinta River is among the principal tropical fluvial systems of North America in terms of discharge and length. According to recent findings, the sources of POM transported along the river change seasonally: there is abundant allochthonous POM input during the rainy season, which changes to an elevated autochthonous production in the dry season [34, 35]. These sources also vary according to the topography: the middle basin's high relief and steep slopes promote significant allochthonous POM input by runoff, whereas the lower slope of the lowland river reaches favors autochthonous POM production [35]. As this POM from diverse sources breaks down, the composition of DOM must inevitably shift during fluvial transport [31, 36].

Through laboratory experiments, the present study aims to elucidate the effect of the seasonal and spatial variations of elemental DOM composition and inorganic nutrients and particulate organic carbon (POC) concentrations on the potential C mineralization and chemical transformation of DOM by the native bacterioplankton communities of the Usumacinta River. We aim to investigate the relationship of the potential C mineralization with the elemental composition and concentration of DOM and the concentration of inorganic nutrients among middle, transition, and lower sites of the river basin and in dry *vs.* rainy seasons. We hypothesize that the potential C mineralization would 1) relate positively to the stoichiometric imbalance of DOM with a greater proportion of C compared to N and P since the microbes adjust their metabolism for the use of non–limiting C (i.e., low C use efficiency) to obtain N and P (e.g., by enzyme production), 2) increase in the middle site because of the higher proportion of C in the DOM associated with allochthonous inputs from the forested landscape, and 3) increase in the rainy season due to allochthonous DOM input derived from the basin. By identifying the factors influencing microbial C mineralization along the Usumacinta River, we contribute to a better understanding of the C dynamics involved in DOM processing within a tropical riverine ecosystem.

## Materials and methods

### Site description

The Usumacinta River originates in the Guatemalan mountain region (upper basin), from where it flows via the middle and lower basins from the southern Mexican border with Guatemala northward before discharging into the south of the Gulf of Mexico (Fig 1). The Usumacinta River basin covers around 56,000 km$^2$. The river's total length within the Mexican territory is 1,100 km [37]. The neighboring Grijalva River joins the Usumacinta 15 km before the river mouth. The average joint discharge is about 2,678 m$^3$ s$^{-1}$.

The basin is characterized by a tropical wet climate with annual average temperatures and precipitation of 8–12°C and 5,000 mm, and 26–30°C and 1,500 mm in the mountainous upper and lowland lower basins, respectively [40]. From June to October, the rainy season presents extreme weather events such as tropical storms and cyclones, resulting in precipitation exceeding 2000 mm per month and seasonal flooding in the lowlands. From November to May, the dry season presents rainfall below 1000 mm per month [40]. Nevertheless, there has been an increase in the likelihood of extreme precipitation events in the basin in the last fifty years attributed to climate change [41].

The Usumacinta River basin includes two Central American geomorphological provinces [42]: karst mountains composed of limestone and dolomite (i.e., the "Sierra de Los Cuchumatanes" and "Los Altos de Chiapas") and the coastal lowland plains (a low gradient zone with knolls and hollows) composed of sedimentary rocks with alluvial and lacustrine formations.

Tropical and subtropical evergreen forests cover the mountainous zone, while tropical rainforests cover the lowlands. The middle basin contains the "Selva Lacandona" in the "Montes

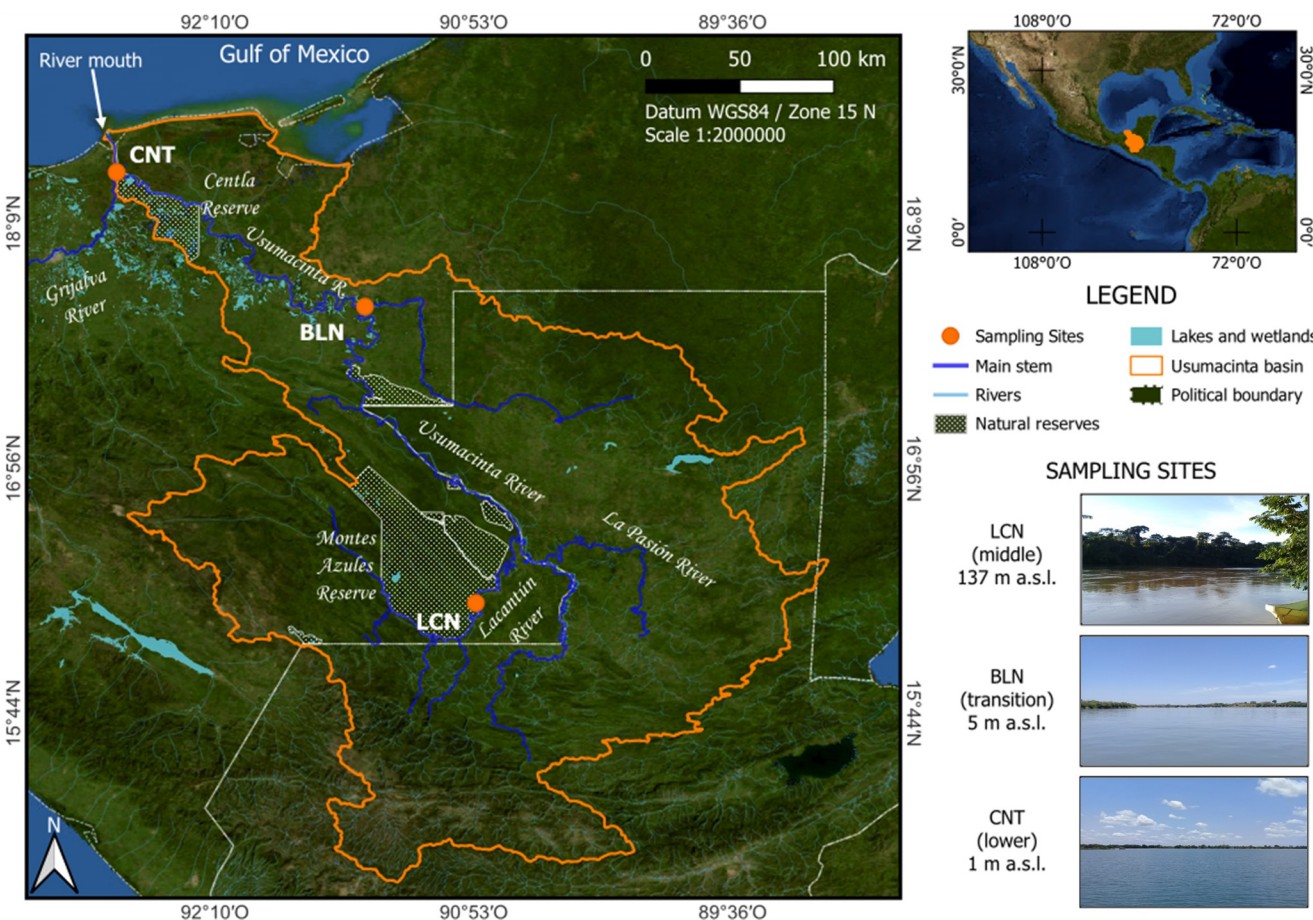

**Fig 1. Sampling sites (Lacantún, LCN; Balancán, BLN; Centla, CNT) at the Usumacinta River basin, southern Mexico.** The basin contour was elaborated using free digital products and services [38, 39]. The U.S. Geological Survey and the National Geospatial Program provide map services and data.

Azules" Biosphere Reserve, the most extensive rainforest in Mesoamerica. One of Mesoamerica's largest and most important wetlands in the lower basin, the "Pantanos de Centla" Biosphere Reserve, extends over 3,028 km². However, agricultural activities and cattle production have caused rainforest losses in both the middle (≈17%) and lower (≈32%) watersheds in recent years [43, 44].

### Field sampling

We performed two field surveys in the Usumacinta River: the first in April–May 2019 (dry season) and the second in October–November 2019 (rainy season). These surveys were conducted during what can be considered typical meteorological periods, according to recent annual and monthly precipitation records in the region [45]. Precipitation in 2019 was 1,767 mm, for an interquartile range (IQR) of 1,653–2,175 in the last 35 years. This year presented monthly rates of 63 ± 63 mm in the dry season (for an IQR of 32–96 mm) and 265 ± 82 mm in the rainy season (for an IQR of 220–343 mm).

Three sampling sites were selected along the middle and lower basins of the Mexican reach of the river (Fig 1 and Table 1): (1) an upriver site in the Lacantún River (Lacantún; 137 m a.s. l.) located within the "Montes Azules" Biosphere Reserve in the middle basin; (2) an intermediate site in the Usumacinta River near the town of Balancán (Balancán; 5 m a.s.l.) at the

**Table 1. Physical and land–cover features of the sampling sites of the Usumacinta River.**

| Characteristics (units) | Lacantún (Middle basin) | Balancán (Transition) | Centla (Lower basin) |
|---|---|---|---|
| Localization (lat/long) | 16°16'40 "N 90°52'18 "W | 17°45'24 "N 91°25'31 "W | 18°24'31 "N 92°38'58 "W |
| Distance (km) [a] | 656 | 286 | 22 |
| Elevation (m a.s.l.) [b] | 137 | 5 | 1 |
| Drainage basin (km$^2$) [b] | 15,772 | 66,623 | 127,736 |
| Maximum depth (m) [c] | 4–5 | 5–20 | 5–15 |
| Current velocity (m s$^{-1}$) [c] | 0.7 ± 0.4–1.4 ± 0.1 | 0.6 ± 0.2–1.1 ± 0.2 | 0.3 ± 0.0–0.8 ± 0.1 |
| Water discharge (m$^3$ s$^{-1}$) [c] | 1 5–173 | 1,080–5,133 | 1,048–5,380 |
| Water temperature (°C) | 22.4 ± 0.0–27.5 ± 0.1 | 25.2 ± 0.0–28.4 ± 0.1 | 27.4 ± 0.4–28.1 ± 1.0 |
| Dissolved oxygen (mg l$^{-1}$) | 8.4 ± 0.0–9.4 ± 0.0 | 6.5 ± 0.0–8.3 ± 0.1 | 2.8 ± 1.1–5.8 ± 1.6 |
| pH | 8.1 ± 0.0–8.0 ± 0.0 | 7.7 ± 0.0–8.0 ± 0.0 | 7.5 ± 0.1–7.8 ± 0.1 |
| Soils [d] | Clay with base saturation and high erodibility (Luvisol, Vertisol, Phaeozem, and Cambisol) | Hydric soil portion (Gleysol, Vertisol, and Fluvisol) | Hydric soil portion (Gleysol, Vertisol, and Fluvisol) |
| Basin's forested area [e] | 89% | 69.8% | 64.7% |
| Basin's cropland area [e] | 8.8% | 26% | 25.8% |
| Basin's wetland area [e] | 0.1% | 0.5% | 1.1% |

[a] Distance from the Usumacinta River mouth at the urban area of Frontera

[b] computation with digital elevation models from the USGS dataset; m a.s.l.: meters above sea level

[c] average, and standard deviation values from the water column in the dry and rainy seasons [34]

[d] edaphology database according to World Reference Base for Soil Resources [46], and

[e] computation with Copernicus Global Land Service dataset (PROBA–V satellite multispectral imagery; [47]; forests encompasses open and closed forests mixing evergreen and deciduous vegetation.

transition point between the high relief (middle basin) and the meander zone (lower basin); and (3) a downriver site (Centla; 1 m a.s.l.) located at the confluence of the rivers Usumacinta and Grijalva in the "Pantanos de Centla" Biosphere within the lower basin. In contrast to the Usumacinta River, the Grijalva River features four dams (La Angostura, Chicoasén, Malpaso, and Peñitas) along its course, as well as the presence of anthropogenic activities with a high impact on water conditions (i.e., pastures, cropland, and urban development). No permits for scientific fieldwork in non-natural protected areas (e.g., Usumacinta River) were required.

Samples were obtained by collecting superficial water (0.5 m of depth) from three points on a transversal cross-section of the river at each site (one center point and two more equidistant from both sides to the center) [35, 48]. The samples were mixed into a composed sample and used to fill 2 L sterile plastic containers to the brim. These containers were stored in darkness at 4°C to minimize sample deterioration until subsequent physical and chemical analyses and incubation experiments could be conducted at the laboratory.

## Experimental design

The potential uptake and mineralization of dissolved C and N mediated by the autochthonous heterotrophic bacterioplankton assemblage from each river site were evaluated in the laboratory through incubation experiments in darkness and under controlled temperatures. Consistent with similar fluvial experiments, the incubations were prepared and initiated within 12 days of sample collection from the river. The laboratory incubations lasted five days [17, 21]. We used the same conditions and periods across all samples to ensure comparability among treatments (i.e., sites and seasons). The river water samples from all sites were divided into two batches. One batch was characterized by immediately analyzing physical and chemical

parameters ($\approx$ 2 hours; see S1 Fig and S1 Table) before incubation. These characteristics were considered the initial reference values ($T_0$). The same physical and chemical variables were analyzed in the second batch immediately after incubation ($T_{120}$; S1 Fig).

The incubations were conducted in closed systems, using 1 L wide–mouth glass jars. We placed a plastic vessel inside each jar containing 100 ml of raw and unfiltered river water from samples and a glass vial filled with 25 ml of 0.1 N NaOH as a $CO_2$ alkali trap for C mineralization measurement (S1 Fig). Five glass sample jars were used per site (5 jars x 3 sites). Furthermore, three jars containing 100 ml of distilled water (instead of samples) and three jars containing only alkali traps were used simultaneously as controls and blanks, respectively. After the plastic vessels and alkali traps were prepared, the glass jars were sealed with airtight lids to prevent the reaction of alkalis with $CO_2$ in the air (i.e., atmospheric contamination).

The glass jars were randomly arranged in an oven and incubated at 25˚C to discard the dependency of the enzymatic activity on the specific temperatures of each site and season. This temperature falls in the range commonly used for microbial incubations and matches the mid-range of the water temperatures previously reported for the river [34]. After 60 h of incubation, the samples were withdrawn from the glass jars and stirred on a magnetic stirrer plate (250 rpm / 2 min) to prevent oxygen depletion. The alkalis were quickly, carefully removed, and sealed during stirring to avoid atmospheric $CO_2$ contamination. After mixing, the vessels and alkalis were incubated for a further 60 h, for a total incubation period of 120 h. The duration of this period and the stirring at the halfway point allowed the maintenance of aerobic conditions throughout the incubations [49].

Temperature, dissolved oxygen concentration (DO), and pH were measured at $T_0$, at the point of stirring, and at $T_{120}$ as experimental controls to identify the physical and chemical changes affecting mineralization. Temperature and DO were measured with a luminescent Hach HQ40D probe, and pH was measured with a digital meter equipped with a glass electrode (Thermo Scientific, Orion 5 Star). Throughout the incubation, the temperature was similar among sites (23.4–24.8˚C), DO remained above 5.9 mg l$^{-1}$, and pH increased slightly over the period (differences from 0.2 to 0.9). These records indicated minor chemical and physical changes and an oxygen demand that did not affect microbial metabolic activity by anaerobiosis.

## Analytical methods in the laboratory

**Particulate fraction.** *Concentration of total suspended solids (TSS)*. TSS evaluation followed the 2540D method [50]. The water samples (400–500 ml) were filtered in pre-weighed Whatman GF/F glass microfiber filters (nominal pore size 0.7 μm). After filtration, the filters were oven-dried (105˚C/48 h) and weighed. The TSS concentration was determined by subtracting the filter weight before filtration from that measured after filtration and dividing the result by the volume filtered.

*Concentration of particulate organic carbon*. For POC analysis, 400 and 500 ml of water samples were filtered into pre–combusted (550˚C/4 h) Whatman GF/F glass fiber filters (nominal pore size 0.7 μm) to retain particulate organic matter. The filters were then acidified with 0.1 N HCl and oven–dried (105˚C/48 h) to eliminate traces of inorganic C (e.g., carbonates). The POC concentration in the filters was determined with a Total Carbon Analyzer (TOC CM 5012, module for solids) by combustion and coulometric detection [51].

*Organic carbon in suspended solids (%OC)*. We calculated the percentage of POC in the TSS as %OC = (POC÷TSS) × 100. A high %OC value indicated a higher organic proportion in suspended solids expected in fresh POM, such as recent debris of vascular vegetation or phytoplankton [52].

**Dissolved fraction.** Water samples were filtered through Whatman GF/F glass microfiber filters (nominal pore size 0.7 μm) to retain and separate the particulate from the dissolved fraction.

*Concentration of total dissolved carbon (TDC) and dissolved inorganic carbon (DIC).* The concentrations of TDC and DIC were determined in water samples by combustion and coulometric detection with a Total Carbon Analyzer (Thermo Scientific, UIC Mod. CM 5012) [51]. We used a liquid module (UIC–Coulometric) for TDC and an acidification module (CM5130) for DIC. The dissolved organic carbon (DOC) concentration was calculated by subtracting DIC from TDC concentrations.

*Concentration of total dissolved nitrogen (TDN), total dissolved phosphorous (TDP), dissolved inorganic nitrogen (DIN), and dissolved inorganic phosphorous (referred to as soluble reactive phosphorus, SRP).* Aliquots (10 ml) of the filtrates were acid–digested at 360˚C with $H_2SO_4$, $H_2O_2$, $K_2SO_4$, and $CuSO_4$. After acid digestion, the Kjeldahl method was used to determine TDN, and the molybdate–ascorbic acid reduction method was used to determine TDP [53, 54]. The extractants were then measured by colorimetry in a Bran Luebbe AutoAnalyzer III. Inorganic forms of N and P were estimated from the water samples before acid digestion to measure the inorganic nutrients. The phenol–hypochlorite method determined DIN concentration ($NO_3^-$ and $NH_4^+$) by colorimetric analysis in a Bran Luebbe AutoAnalyzer III [55]. We used the molybdate–ascorbic acid method to measure the SRP concentration by colorimetric analysis in a Bran Luebbe AutoAnalyzer III [53].

*Concentration of dissolved organic nitrogen (DON) and dissolved organic phosphorus (DOP).* The DON and DOP were calculated by subtracting the inorganic forms from the total dissolved forms as DON = TDN − DIN and DOP = TDP − SRP.

**Potential carbon mineralization.** Following the hydrochloric acid titration method [49], potential C mineralization was determined by the C emissions rate as $CO_2$ (C–$CO_2$) in the incubations. The chemical balance indicated that each mole of $CO_2$ released through microbial mineralization was chemically trapped as sodium carbonate by reacting with two moles of alkali (NaOH; Eq 1). After incubation, 1.5 N $BaCl_2$ was added into the alkali traps to precipitate the sodium carbonate into insoluble $BaCO_3$ (Eq 2), as follows:

$$2\, NaOH + CO_2 \rightarrow Na_2CO_3 + H_2O \tag{Eq1}$$

$$BaCl_2 + Na_2CO_3 \rightarrow BaCO_3 + 2NaCl \tag{Eq2}$$

Alkali solutions with precipitates were then titrated with 0.1 N HCl using phenolphthalein as an indicator:

$$NaOH + HCl \rightarrow NaCl + H_2O \tag{Eq3}$$

The volume of acid needed in the titration indicated the quantity of alkali, which did not react with $CO_2$. Thus, the reacted quantity (non–reacted and initial alkali difference) represented half of the $CO_2$ emitted from samples (Eq 1 and Eq 4). Finally, the C flux from the emitted $CO_2$ (C–$CO_2$) was calculated as follows:

$$CO_2 - C = \frac{(V_S - V_B) \times A_r C}{2} \times V_{WS} \div T_{INC} \tag{Eq4}$$

Where $V_S$ is the volume of HCl (ml) consumed by the alkali in the titration, $V_B$ is the volume of HCl consumed by the alkali of blanks. $A_r$C is the relative atomic mass of carbon, $V_{WS}$ is the volume of the water sample (ml), and $T_{INC}$ is the incubation period (days). The C–$CO_2$ fluxes average in controls ($10.2 \pm 2.5$ mg $l^{-1}$ day$^{-1}$ in the dry season and $8.4 \pm 3.4$ mg $l^{-1}$ day$^{-1}$

in the rainy season) were subtracted from those for samples due to the atmospheric $CO_2$ that could enter the water before the incubations considering the solubility of $CO_2$ [56] and the water acidification [57].

## Data treatment and statistical analyses

The C and N transformations between the organic and inorganic pools were evaluated by the differences in dissolved C, N, and P between points $T_0$ and $T_{120}$, represented by the delta value ($\Delta$). For example, positive $NO_3^-$ and $NH_4^+$ deltas indicated inorganic N mineralization, while negative deltas evidenced inorganic N uptake. Likewise, the changes in DOC and DON between points $T_0$ and $T_{120}$ ($\Delta DOC$ and $\Delta DON$, respectively) represented the C and N taken up by microbes from the DOM in the batch incubations [58]. Relative changes of the dissolved C, N, and P concentrations after incubation ($\Delta\%$) were also measured as the concentration change divided by the initial reference value ($T_0$) and expressed as percentages for each variable.

A Principal Component Analysis (PCA) was applied using the dissolved fraction concentrations at $T_0$ to summarize the DOM variation and explore the initial chemical characteristics of the samples. The PCA scores were correlated with TSS, POC, and %OC to identify environmental patterns (e.g., river discharge, basin erosion, and primary production) and DOM supply among seasons and sites. The PCA employed Box-Cox transformations (with $\lambda = 0.75$) to normalize the data, as well as an Euclidean similarity index, and a data correlation matrix to ensure normality, linearity, and standardization in the ordination process. The limit of detection (LOD) for N and P, set at 0.1 and 0.3 mg l$^{-1}$, respectively, was applied within the PCA for values falling below these thresholds. A subsequent PCA focused on the dissolved fraction differences, utilizing a correlation matrix and an iterative imputation method to account for missing values [59]. This analysis linked the components identified in the first PCA and the carbon dioxide ($C$–$CO_2$) production rate with those derived from the second PCA. For both PCAs, significant components were selected based on expected eigenvalues using a random model approach (Broken Stick method).

We incorporated a linear mixed–effects model to evaluate the influence of C/N/P ratios in the DOM on the $C$–$CO_2$ production rate. This model treated seasons, sites, and C/N/P ratios as fixed effects while considering the sample bottle as a random effect to adjust for sample interdependence. We used linear regression models to explore the dissolved fraction variables driving the $C$–$CO_2$ rates at $T_0$, and a stepwise regression model was specially performed on delta values to identify the changes in dissolved nutrients explaining the $C$–$CO_2$ rate variation. We used the second PCA and the regression models to assess the first hypothesis relating C mineralization to elemental DOM composition.

To assess the second and third hypotheses about spatial and seasonal differences of the potential C mineralization, the DOM deltas and $C$–$CO_2$ rates were compared between two factors: season with two levels (dry and rainy) and site with three levels (Lacantún—middle—, Balancán—transition—, and Centla—lower—sites). These comparisons were made using a mixed–design ANOVA to correct for the spatial and seasonal dependency in the samples. The PCAs and the comparisons of factor levels were implemented in the statistical software Sigma-Plot (v14), Past (v4.08; packages: lme4, MASS, and rstatix), and R (v 4.3.1; R Core Team, 2019). A p-value of 0.05 was used as the threshold for significance in all the tests.

## Results

### Initial conditions ($T_0$)

Nutrient concentrations, especially dissolved fractions, varied more among sites in dry than rainy seasons (S2 Table). In the dry season, the DOC in Lacantún and Balancán (representing

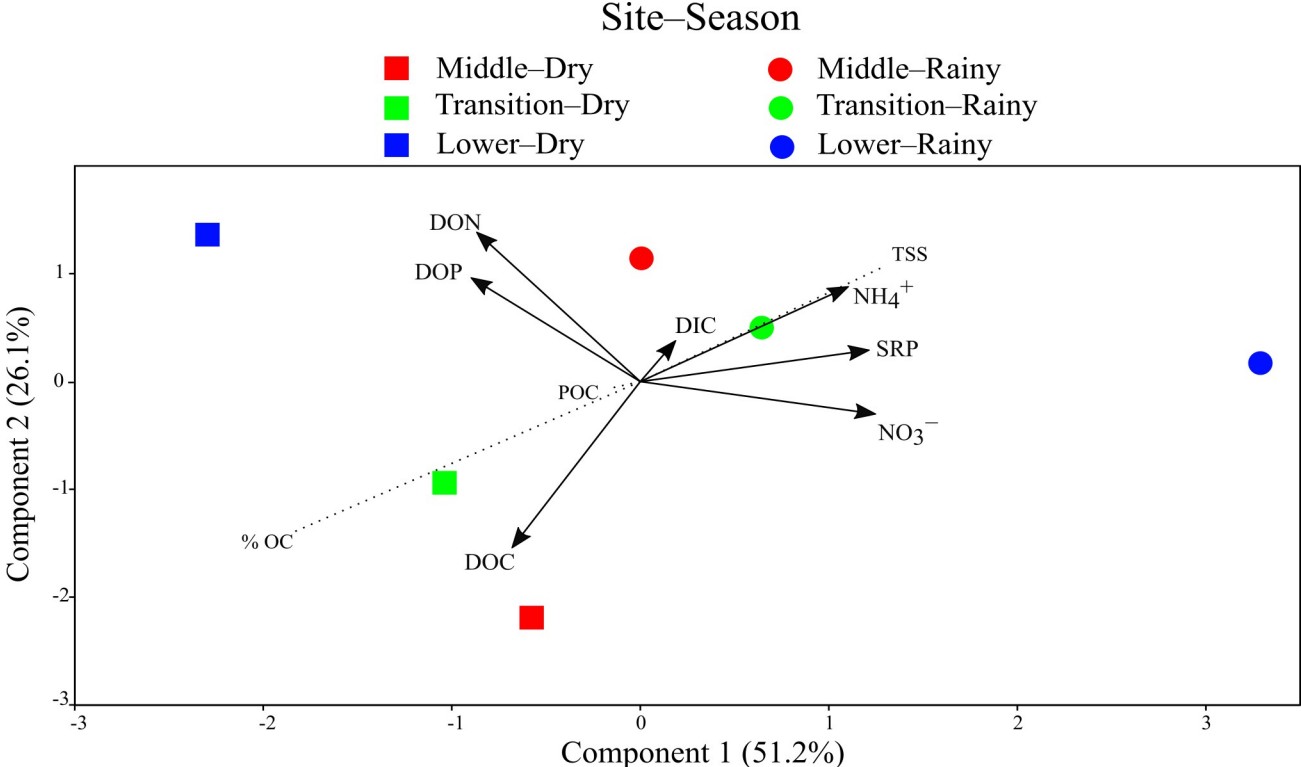

**Fig 2. PCA of the dissolved nutrients in the Usumacinta River water samples at T0.** The dissolved fraction loadings are indicated as arrows, and the correlation of particulate variables with ordination scores is shown as dotted lines. Acronyms are given in the S1 Table.

the middle and transitional sites: $\approx$ 21,000 µg l$^{-1}$) was roughly double that of Centla (lower site: 12,620 µg l$^{-1}$). Moreover, the NO$_3^-$ decreased downriver, dropping below the LOD in Centla, where the NH$_4^+$ peaked. The SRP was below the LOD. Downstream, the DOC/DON and DOC/DOP ratios declined, a trend attributed to the rising concentrations of DON and DOP with a decrease in DOC. In the rainy season, DOC variation was negligible (7,260–8,650 µg l$^{-1}$), while NO$_3^-$, NH$_4^+$, and SRP increased downstream. The DOC/DON and DOC/DOP ratios increased downstream as DON and DOP decreased.

The PCA in Fig 2 shows that the rainy season observations, with high NO$_3^-$, NH$_4^+$, and SRP concentrations, are separated from the dry season observations, with high DOC, DON, and DOP concentrations in the first component. This component positively correlates with TSS (to the rainy season) and negatively with %OC (to the dry season; r = 0.52 and −0.75, respectively). The second component distinguishes Lacantún and Balancán in the dry season, with high DOC, from other observations with elevated concentrations of DON, DOP, and NH$_4^+$ (Fig 2). The DOC/DON and DOC/DOP ratios were generally high in the dry season.

## Post–incubation (T$_{120}$)

In the dry season, the ΔDOC was considerable, with lower values in the Lacantún and Balancán sites (middle and transition zones) (−14,220 ± 1,110 µg l$^{-1}$ and −13,520 ± 2,160 µg l$^{-1}$, respectively) than in Centla (−9,270 ± 1,790 µg l$^{-1}$; t(4) = −5.7 and t(4) = −3.2, respectively, p < 0.01; Fig 3). This steep reduction in C in Lacantún and Balancán and a slight rise of N and P caused a decline in the DOC/DON and DOC/DOP ratios throughout the incubation (S2 Table). Moreover, Lacantún also had high ΔNH$_4^+$ (18 ± 8 µg l$^{-1}$) than Balancán and Centla

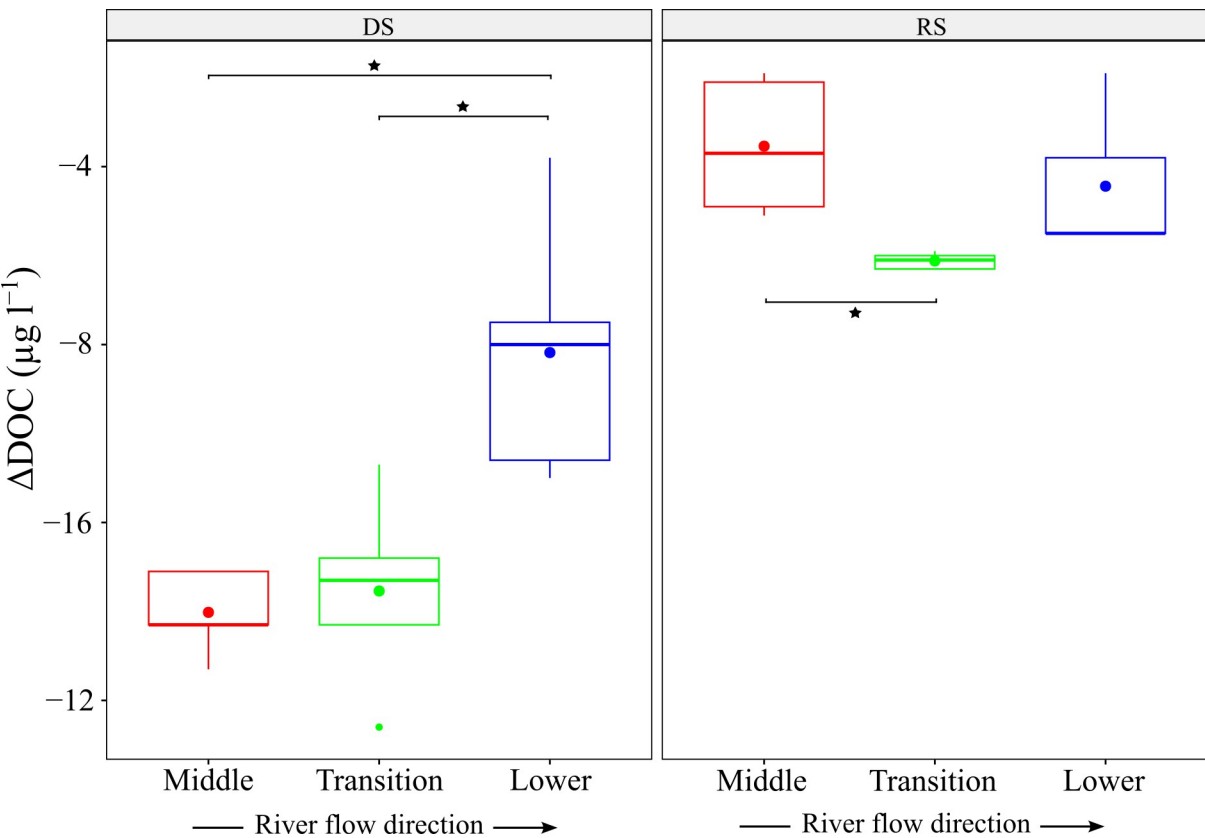

**Fig 3. DOC consumption at $T_{120}$ of the water samples incubations.** DS: dry season; RS: rainy season. The dots represent the average. The horizontal brackets represent significant differences among groups (see main text).

(nearly 6 µg l$^{-1}$). In the rainy season, the ΔDOC was similar among sites, with values from −6,110 to −3,950 µg l$^{-1}$, though Lacantún had a higher delta than Balancán (t(4) = 3.65, p = 0.02; Fig 3). The ΔNO$_3^-$ showed decreases from 4 to 29 µg l$^{-1}$, and the ΔNH$_4^+$ averaged −102 ± 40 µg l$^{-1}$. The DOC/DON and DOC/DOP ratios decreased after the rainy season incubation due to a loss in C and a gain in N (S2 Table).

After the incubation, the PCA of deltas in the dissolved fraction presented a first component that was positively correlated with ΔDOC, ΔDON, and ΔDIC towards the rainy season samples but negatively correlated with ΔNO$_3^-$ and ΔNH$_4^+$ towards the dry season samples, especially separating Lacantún and Balancán (X-axis scores: < −1.5 in Fig 4); in particular, the ΔDOC in the dry season was significantly lower than in the rainy season (F(1,4) = 100.8, p < 0.01). This component was also positively correlated with the first and second components of the PCA at $T_0$ and negatively with the C–CO$_2$ rates. The second component had a positive correlation with ΔNO$_3^-$, ΔDON, and ΔDOP and a negative correlation with ΔDIC and ΔNH$_4^+$ separating the samples from Lacantún and Balancán from Centla. Linear regression models showed that the variation of the mean values of the DOC/DON ratio at $T_{120}$ and the ΔDOC/ DON were directly related to the DOC/DON at $T_0$ (S3 Table). The DOC/DOP at $T_0$ is also directly associated with ΔDOC/DOP.

## C–CO$_2$ rates

The C–CO$_2$ rates of sites ranged between 5.5 and 14.2 mg l$^{-1}$ day$^{-1}$ on average (Fig 5). The ANOVA indicated a significant effect of season on C–CO$_2$ rates (F(1, 4) = 8.5, p < 0.02),

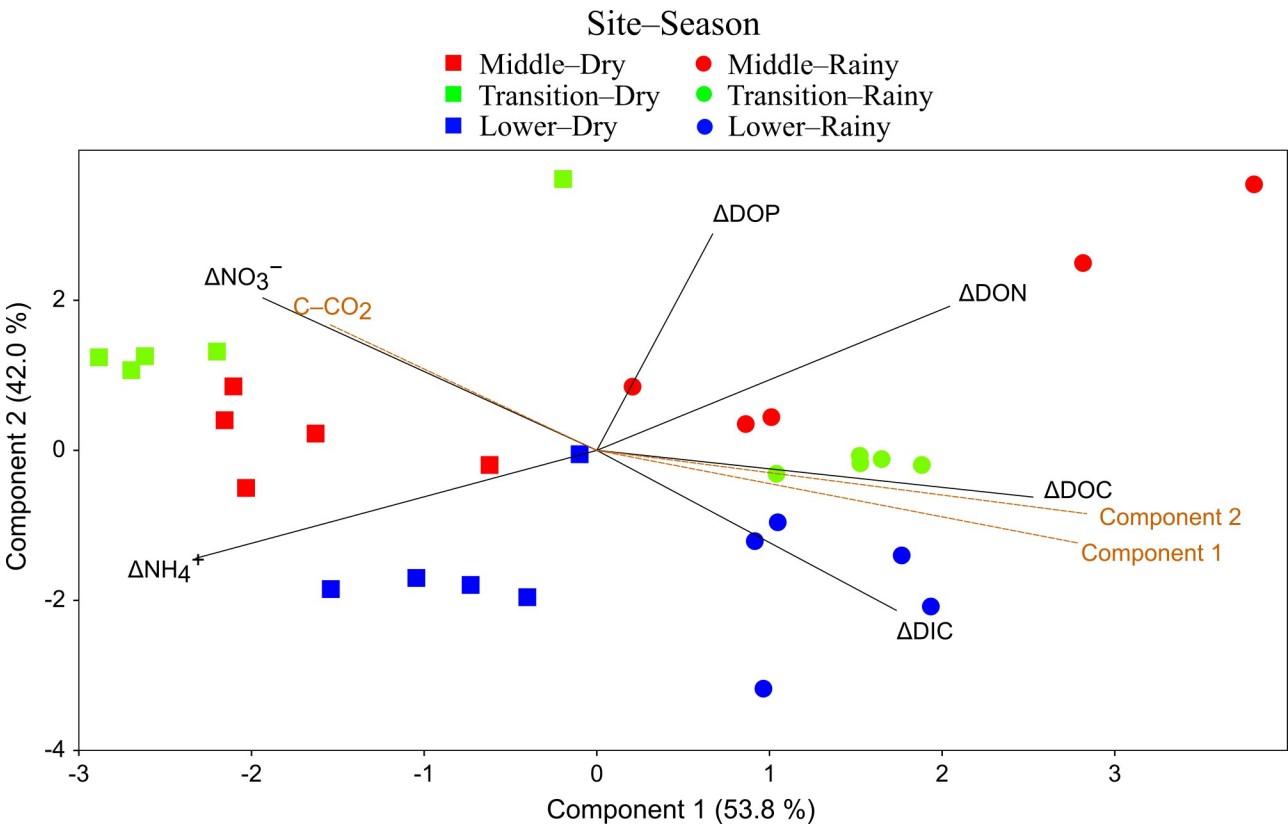

**Fig 4. PCA of the dissolved nutrients' deltas at $T_{120}$ in the water samples incubations.** Straight lines indicate the loadings of the variables. The dotted lines are correlations of the components with the $CO_2$-C rate and the components of the PCA at the initial incubation condition ($T_0$). Acronyms are given in the S1 Table.

which were higher in the dry than in the rainy season (p = 0.03). In the dry season, the C–$CO_2$ rates were higher in Lacantún and Balancán (12.0 ± 4.5 and 14.8 ± 2.7 mg l$^{-1}$ day$^{-1}$, respectively) than in Centla (6.3 ± 0.7 mg C–$CO_2$ l$^{-1}$ day$^{-1}$; t(4) = 3.1 and t(4) = 7.1, p < 0.05; Fig 5 and S4 Table). Conversely, no significant differences in C–$CO_2$ rates were found among sites in the rainy season (F (2,8) = 0.2, p = 0.81).

Based on linear regression models, the concentration of DOC at $T_0$ explained 67% of the variation in C–$CO_2$ rates, with higher rates corresponding to higher DOC concentration (Table 2). After the incubation ($T_{120}$), the C–$CO_2$ rates increased with the DOC/DOP ratio, but the DOC/DON and DON/DOP ratios showed no effect on the C–$CO_2$ rates (Fig 6). Based on stepwise regression models using the deltas in the dissolved fraction, the ΔDOC and ΔNH$_4^+$ were the variables that best explained the C–$CO_2$ rates variation (Table 2).

## Discussion

### Relationship between elemental ratios in DOM and microbial C mineralization

The highest C–$CO_2$ rates were observed in the samples with high initial DOC concentration (Fig 4 and Table 2), suggesting that organic C availability boosts potential C mineralization. Nevertheless, the low availability of dissolved nutrients could also impact microbial activity and cause high C–$CO_2$ rates since the high DOC concentration coincided with high DOC/

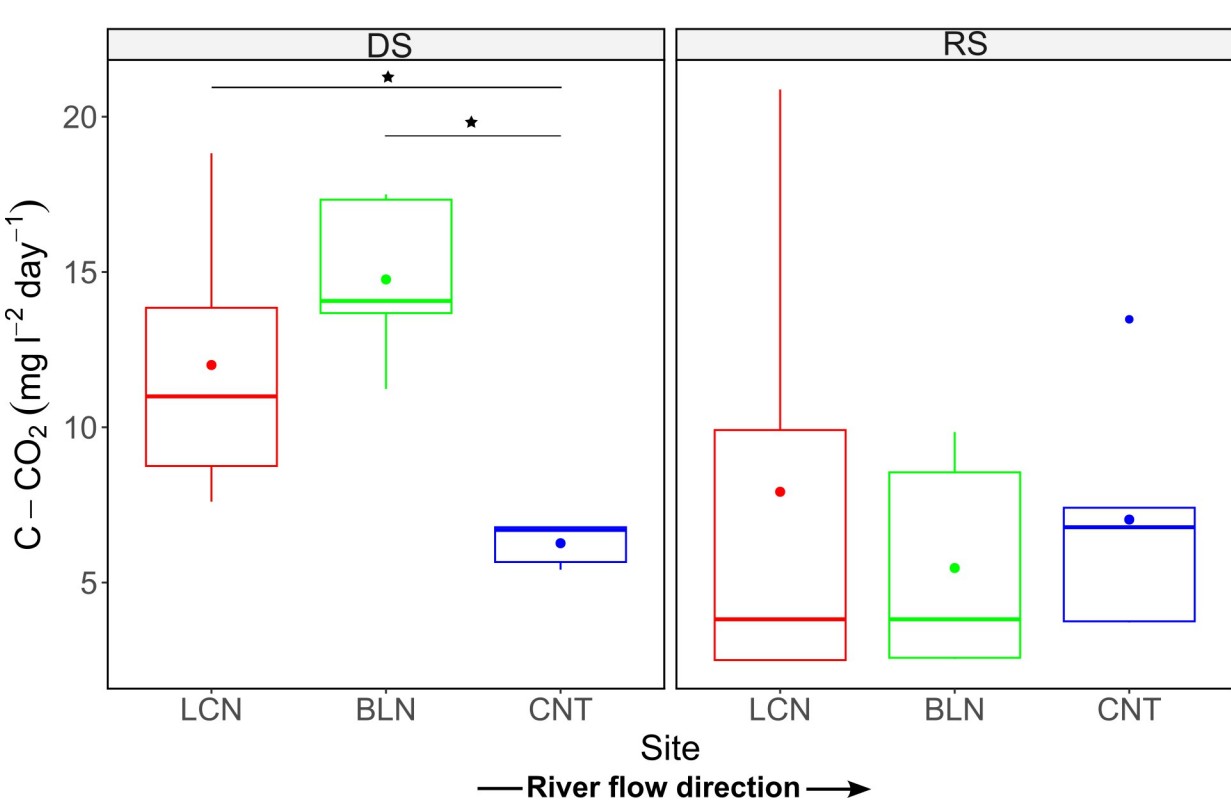

**Fig 5. Variation of the potential carbon mineralization ($C–CO_2$) at $T_{120}$ in the water samples incubations.** DS: dry season; RS: rainy season. The dots represent the average. The horizontal brackets represent significant differences between groups (see main text).

DON and DON/DOP ratios (Fig 2). Indeed, according to the linear mixed models, the $C–CO_2$ rates increased in line with the DOC/DOP ratio (Fig 6). This agrees with the first hypothesis that proposed a relationship between the elemental DOM composition and C mineralization. Similar studies have also identified elevated respiration rates in organic substrates with a surplus of C and scarcity of P and N through microbial experiments that involve culturing and incubating heterotrophic aquatic bacteria [60, 61]. These increased rates are expected to result in heightened C mineralization, which aligns with the outcomes observed in our study.

A high proportion of C in DOM with high mineralization rates might indicate that microbes invest more energy in mineralization to obtain limited nutrients. Sinsabaugh et al. [62] reported that carbon use efficiency (CUE) in the microbial community decreases with an increase in the C/N and C/P ratios of the resource. These authors defined CUE as the ratio of C assigned for growth to the C assigned for respiration. Therefore, if the CUE has low values,

**Table 2. Stepwise regression models for the $C–CO_2$ rates at the initial ($T_0$) and final ($T_{120}$; delta values) conditions of incubations.**

| Condition | Model | df | $r^2$ | F | P value |
|---|---|---|---|---|---|
| $T_0$ | $C–CO_2$ = **0.0004(DOC)** | 1 and 4 | 0.67 | 8.1 | 0.04 |
| $T_{120}$ | $C–CO_2$ = **−0.07(ΔDOC)** + 0.02(ΔNO$_3^-$) **−0.006(ΔNH$_4^+$)** | 3 and 25 | 0.40 | 7.3 | <0.01 |

The $T_0$ model used a simple linear regression model. $T_{120}$ model used backward stepwise with the following predictors: ΔDIC, ΔDOC, ΔDON, ΔNO$_3^-$, ΔNH$_4^+$, ΔDOP. The bold indicates significative explanatory variables (p < 0.05).

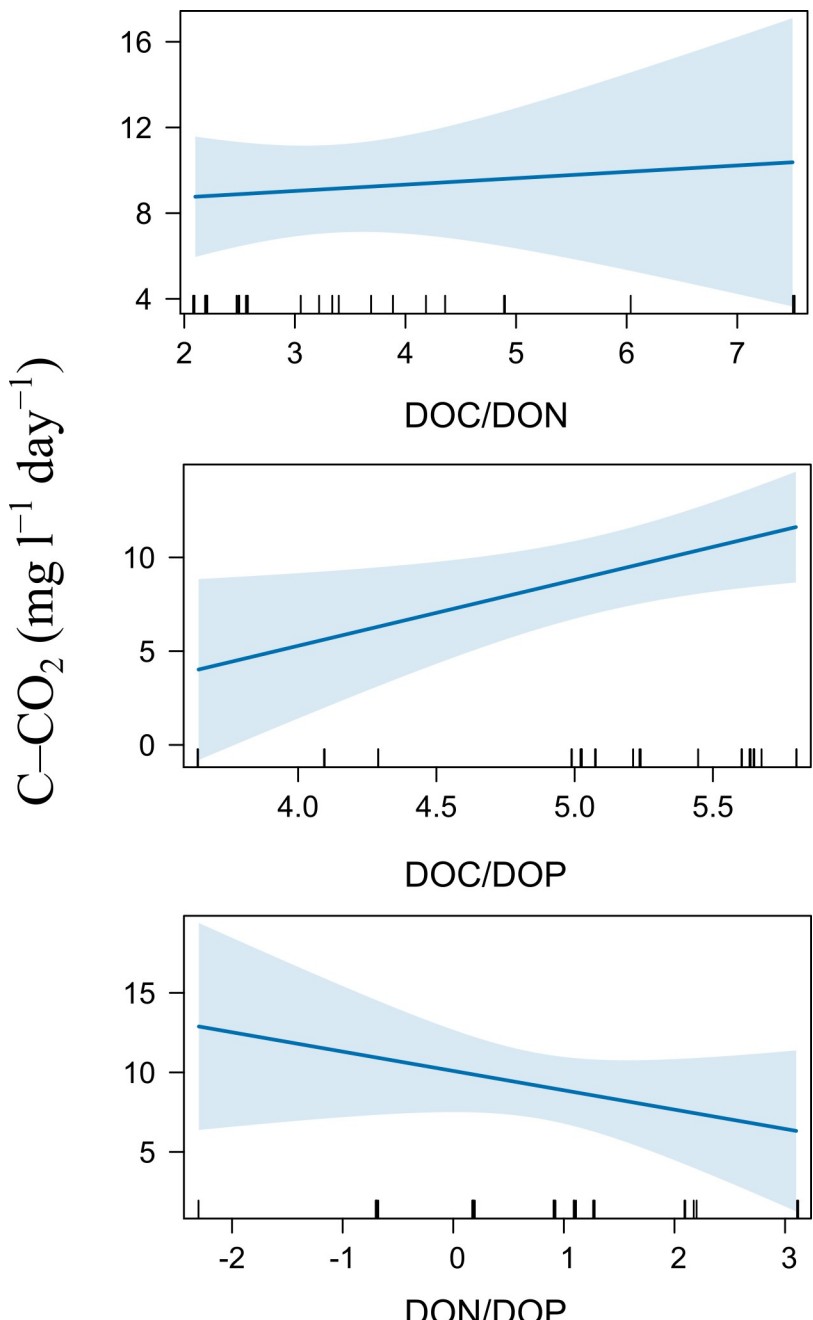

**Fig 6. Effect of elemental DOM composition (C/N/P ratios) on C–CO$_2$ rate after water samples incubations ($T_{120}$).** The blue line represents a mixed–effects model, and the blue area represents its 95% confidence interval.

the microbial community invests more C in basal metabolism associated with producing bio-molecules as enzymes. Consequently, the microbial community must invest energy to produce extracellular enzymes to break down organic molecules when the DOM has high C/N and C/P ratios [63, 64]. Other experimental studies have found constraints in microbial processing related to elemental DOM composition in field and laboratory data for lotic environments, such as high bacterial growth efficiency under low DOC/DON and DOC/DOP ratios [65, 66]. The high C mineralization rates in our results could represent a strategy for fitness

improvement by the microorganisms (e.g., exoenzyme production), indicated by a low efficiency of use of C when the DOM has high C/N and C/P ratios.

Incubation resulted in a decrease in the DOC/DON and DOC/DOP ratios. Higher initial ratios led to greater decreases. This decrease occurred due to a decline in C and a slight gain in N and P. Other studies have observed an increase in DOC/DON and DOC/DOP ratios during incubations of river water samples [30] or high water residence time [67] since the microbes mineralize N and P to obtain limited inorganic nutrients from the dissolved organic pool. However, the decrease in the resources' DOC/DON and DOC/DOP ratios has also been reported after microbial culturing [23] or dark incubation [68] elsewhere, especially when DOM was rich in C [68]. This decrease was attributed to higher microbial respiration and aligns with the match between the DOC/DON and DOC/DOP ratios decrease and the direct relationship between the C–$CO_2$ rate and the DOC/DOP ratio shown in this study. By the CUE ratio, the microbes must have consumed and mineralized more C than N and P during incubation. Since the intracellular microbial composition usually had a high N and P proportion, the preferential processing of C indicates that the microorganisms utilized surplus C to cope with the low availability of N and P. Therefore, the imbalance of C concerning N and P within the DOM accounted for the variation in C processing by heterotrophic bacterioplankton in the Usumacinta River.

## Differences in microbial C mineralization among sites

The differences in DOC, DON, and DOP concentrations among the sites at the beginning of the experiment in the dry season reflect shifts in DOM composition and quantity as the landscape elevation and relief change. The high DOC/DON and DOC/DOP ratios at Lacantún (middle site) before incubation indicate that the tropical rainforest supplies considerable amounts of organic matter to the river. Forest vegetation and soils usually export C–rich DOM to rivers because of the high proportion of C in terrestrial plant tissues [11]. For example, Elser et al. [22] reported that terrestrial autotrophs had C/N and C/P ratios three times higher than that of aquatic autotrophs (36 and 10, respectively). Forests also supply low N and P to rivers because of the retention of both within the biota on land [69], as well as the fact that the gaseous N loss rate is higher than the leaching rate (e.g., during denitrification) [70]. Traces of the terrestrial influence remained as far downstream as the beginning of the lower Usumacinta River basin since the proportion of C in the DOM in Balancán (transition site) was also high. In contrast to the forested middle Usumacinta River basin, the agriculture, cattle production, and urban development present in the lower basin release considerable amounts of dissolved N and P into the river [40, 43], as has commonly been reported in other river basins [71, 72].

Our results highlight differences in DOM processing among the different sites in the dry season, which corresponded with stoichiometric differences in the available DOM. The higher values of DOC uptake at Lacantún and Balancán with the high C in elemental ratios imply that the microbes use a significant amount of C when the DOM is derived from terrestrial sources (Figs 3 and 7). This pattern is consistent with previous observations in which the microbial uptake of DOC [73, 74] and the C mineralization [68] increase in headwaters and forested temperate streams, especially in upland C–rich waters. The chemical characteristics of terrestrial–derived DOM have traditionally been viewed as recalcitrant, with high DOC/DON and DOC/DOP ratios indicating a higher content of aromatic and high–weight molecules [66]; However, our study, in line with recent evidence, suggests that the lability of this DOM to bacterial metabolism is contingent on the sources in the landscape. Forests can export organic compounds of low molecular weight, such as carboxylic acids, amino acids, and carbohydrates, that remain following partial biodegradation in soils and favor aquatic microbial DOM uptake

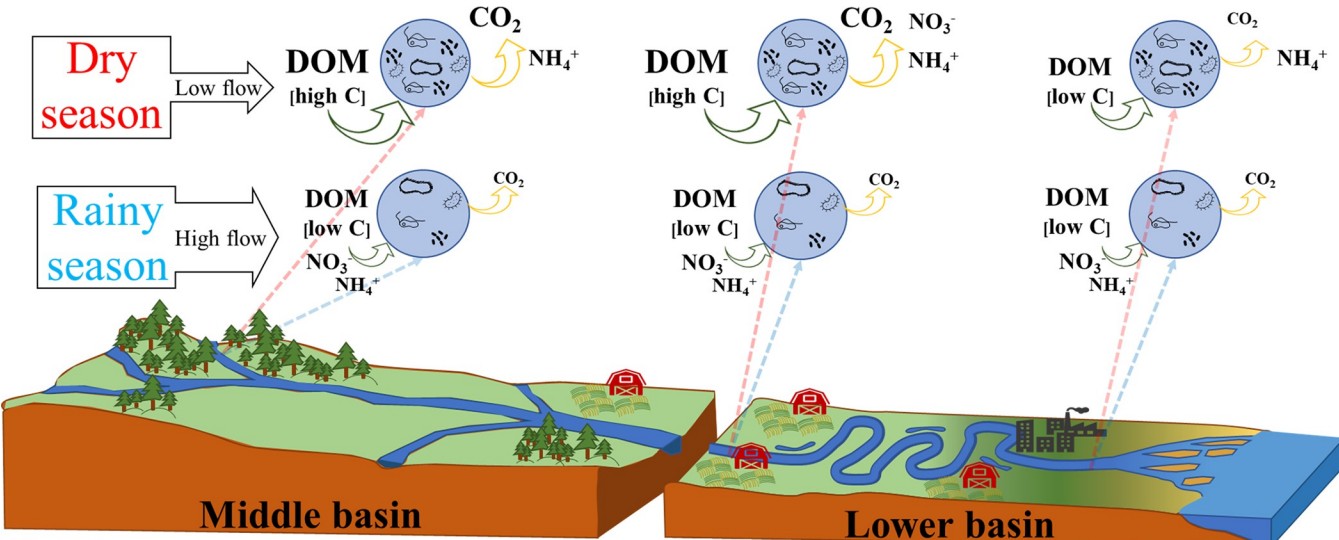

**Fig 7. Linkages of DOM's concentration and elemental composition with potential microbial mineralization along a tropical river system.** In the dry season, highland rainforests export abundant DOM with high C/N and C/P ratios, boosting DOC uptake and C mineralization in the river until reaching the initial section of the lower basin. Lower DOM amount with low C/N and C/P ratios near the river mouth decreases the DOC uptake at a low C mineralization rate. Conversely, the high water flow in the rainy season washes the microbial community out of the water column, decreasing microbial activity.

[73, 75]. Other attributes of the consumed DOM composition could determine whether the microbes allocate C from this DOM to bacterial biomass or respiration.

The higher $C–CO_2$ rates at Lacantún and Balancán in the dry season suggest that the microbes responded to abundant DOM with surplus C by consuming and mineralizing a significant part of the DOC from the forested catchments (Figs 3 and 7). The metabolic strategy of microorganisms can thus change according to the elemental composition of the DOM. Other studies in tropical rivers have found high aquatic heterotrophic respiration driven by DOM inputs from terrestrial environments, such as the high respiration rates at mid-elevation streams during the dry season in forested regions of the Mara River, Kenya [19]. Indeed, planktonic bacteria can increase their catabolism during periods with high terrestrial DOC inputs from tropical rainforests [60]. Respiration rates in the water column of the Amazon River, for instance, rise during the dry season as microbes inefficiently use compounds like lignin and other macromolecules derived from the rainforest [15]. The DOC that arrives from forested catchments can also constitute a substantial C pool, boosting microbial respiration in lowland receiving waters [76], which agrees with the high $C–CO_2$ rate recorded in this study at Balancán. In addition to high C/N and C/P ratios, the low inorganic nutrient concentrations matched the elevated C mineralization rates in Lacantún and Balancán, suggesting it is more expensive for microbes to retain C within biomass under N and P limitations than to utilize it for essential metabolic functions under excessive C availability. Thus, microbes rapidly mineralize the DOC by increasing respiration rates when abundant terrestrial DOM inputs from the middle basin of the Usumacinta River enter the water column.

The similar DOC concentration among the sites in the rainy season mirrors the effect of the high water flow and quick mobilization of DOM along the Usumacinta River towards the sea. The similar concentrations of $\Delta DOC$, $\Delta NO_3^-$, and $\Delta NH_4^+$, along with similar $C–CO_2$ rates, among the sites in the rainy season suggest that the high water flow conditions along the river act to homogenize the activity of microbes in the samples.

## Changes in the microbial C mineralization between seasons

The changes in the particulate fraction variables between the dry and rainy seasons before incubation evidenced different seasonal POM inputs in the Usumacinta River. The low TSS and high %OC (9.2–11.1%) reflected an increase in autochthonous–derived POM in the dry season. This increase is consistent with reports of photoautotroph growth stimulated by decreased turbidity and increased residence time of nutrients in the water column during reduced water flow in the Usumacinta River (e.g., phytoplankton increase) [35, 77]. The seasonal dynamic of the Usumacinta River is similar to that of tropical rivers in Africa and India, where the OC content in suspended sediments increases during low water periods and can range from 5 to 26% [78, 79].

In contrast, the low DOC concentration in the rainy season could represent the effect of dilution in the basin since the water discharge strongly regulates the flux and transport of DOC in the Usumacinta River during that season [48]. Inverse relationships between DOC concentration and water discharge can occur in temperate aquatic systems [80, 81]. The higher TSS and lower %OC (2–3.6%) in the rainy season suggest allochthonous–derived POM entering the Usumacinta River when the river presents elevated water flow and high turbidity. These conditions diminish primary production and autochthonous–derived organic matter.

Although elemental composition explained the spatial variation of C–$CO_2$ rates by evidencing shifts of DOM sources, our results also show that the amount of DOM available for microorganisms impacts the seasonal variation of C mineralization. The decrease in DOC concentration towards the rainy season implies a lower quantity of substrate to metabolize, constraining microbial activity and C mineralization. Similarly, Lynch et al. [81] found a decrease in respiration in microbial communities with low autochthonous productivity and DOC concentrations during periods of high water flow and vice versa. Moreover, the increased water flow in the Usumacinta River during the rainy season could wash microbiota out from the watercourse, decreasing the number of microbes in samples taken during that time (Fig 7). The low residence time in large tropical rivers decreases microbial DOM degradation [82] because the increased water current decreases DOM bioavailability and washes out the planktonic communities [5, 17]. Therefore, with lower DOM concentration and microbial abundance, samples' incubations in the rainy season processed less DOC and DON.

The increased availability of DOM in the dry season explained the differences in microbial processing, which is evident in the second PCA (Fig 4). The microbes consumed more DOC in the dry than in the rainy season, indicating that the ample supply of autochthonous–derived DOM prompted higher microbial retention of DOC. Microbes rapidly consume DOM from leachates of photoautotrophs or after enzymatic cleavage since these compounds, such as carbohydrates and carboxylic acids, favor the assimilation of C for growth. Moreover, high N and P concentrations of photoautotrophs boost microbial degradation [11, 17, 30]. Earlier observations in tropical aquatic systems with marked hydrological seasonality also show higher DOC uptake in the dry season or during periods with phytoplankton production [60, 74].

Moreover, the high availability of DOM with low nutrient availability before incubation also prompted microbial mineralization of N in the dry season. Ammonification usually rises as the availability of C increases, except when nitrification remains stable. This process agrees with the positive delta values of $NO_3^-$ and $NH_4^+$ in the dry season when Lacantún and Centla had higher $\Delta NH_4^+$ but lower $\Delta NO_3^-$, while Balancán showed the opposite effect, possibly due to higher nitrification with a lower concentration of DOC. Other studies indicate that nitrifying bacteria increase their activity when heterotrophic bacteria are C–limited [83, 84]. These results suggest that nitrification is the dominant process of N transformation in the lower basin site during the dry season.

## Conclusions

Our findings indicate that increasing the C/N ratio and mainly the C/P ratio in DOM leads to higher C loss due to mineralization. This effect caused a reduction in the DOC/DON and DOC/DOP ratios as $CO_2$ was released. Differences in elemental DOM composition between the middle and lower basins of the tropical Usumacinta River were the main driver of the C–$CO_2$ rates due to the export of DOM with a high proportion of C from rainforests, as well as autochthonous DOM production (with a low proportion of C) in the river sites surrounded by agricultural land. This pattern only appeared in the dry season when spatial differences in DOM availability arose under low water flow. In the dry season, the particulate fraction (TSS and %OC) evidenced a higher availability of autochthonous–derived sources, corresponding with higher DOC consumption and $NO_3^-$ and $NH_4^+$ production during incubation. In contrast, the high water flow in the rainy season reduced DOM availability in samples, lowering the C–$CO_2$ rates and their variation among sites. Autochthonous–derived DOM seems more closely related to C immobilization into biomass during microbial metabolism than C mineralization. According to the differences in C–$CO_2$ rates, DOC concentration and uptake, and the DOC/DON and DOC/DOP ratios, our study demonstrates that the hydrological regime and longitudinal changes in the topography and land cover of the basin directly influence the DOM characteristics (allochthonous vs. and autochthonous) and indirectly can affect the metabolism of heterotrophic bacterioplankton in the Usumacinta River.

## Supporting information

**S1 Fig. Experimental arrangement to determine the potential C mineralization and N transformation by microbes.** At each season (dry and rainy) and in each site (Lacantún, Balancán, and Centla), one river water sample batch was used to determine initial water conditions (dotted arrow; referred to as $T_0$ in the main text). The remaining sample was used to perform the incubation experiment and determine the final water conditions (straight arrow; referred to as $T_{120}$ in the main text).
(TIF)

**S1 Table. Evaluated physical and chemical variables at the incubation experiment's initial ($T_0$) and final ($T_{120}$) times.**
(DOCX)

**S2 Table. Changes of particulate and dissolved nutrients after incubating water samples from Lacantún, Balancán, and Centla in two contrasting seasons.**
(DOCX)

**S3 Table. Slope and intercept of linear regressions (df: 1 and 4) between C/N/P ratios at $T_{120}$ (log transformed and delta values) and C/N/P ratios at $T_0$ of the dissolved organic matter.**
(DOCX)

**S4 Table. Central tendency values and of the potential C mineralization ($CO_2$–C) after incubation ($T_{120}$ at 25°C) of the water samples from the Usumacinta River sites [Lacantún, Balancán, and Centla] in the dry and rainy seasons.**
(DOCX)

## Acknowledgments

Rodrigo Velázquez assisted with laboratory testing and performed nutrient analyses. Daniela Cortés-Guzmán, Luis A. Oseguera, Ismael Soria-Reinoso, Jorge Ramírez, and Julio Díaz

helped with fieldwork and data collection. The foundation Natura y Ecosistemas Mexicanos AC supported the logistics at the Chajul Biological Station.

## Author Contributions

**Conceptualization:** Daniel Cuevas-Lara, Felipe García-Oliva, Javier Alcocer.

**Data curation:** Daniel Cuevas-Lara.

**Formal analysis:** Daniel Cuevas-Lara, Felipe García-Oliva, Salvador Sánchez-Carrillo, Javier Alcocer.

**Funding acquisition:** Felipe García-Oliva, Javier Alcocer.

**Investigation:** Daniel Cuevas-Lara, Felipe García-Oliva, Salvador Sánchez-Carrillo, Javier Alcocer.

**Methodology:** Daniel Cuevas-Lara, Felipe García-Oliva.

**Project administration:** Felipe García-Oliva, Javier Alcocer.

**Resources:** Felipe García-Oliva, Javier Alcocer.

**Software:** Daniel Cuevas-Lara.

**Supervision:** Felipe García-Oliva, Salvador Sánchez-Carrillo, Javier Alcocer.

**Validation:** Felipe García-Oliva, Salvador Sánchez-Carrillo, Javier Alcocer.

**Visualization:** Daniel Cuevas-Lara, Felipe García-Oliva, Javier Alcocer.

**Writing – original draft:** Daniel Cuevas-Lara, Felipe García-Oliva, Javier Alcocer.

**Writing – review & editing:** Daniel Cuevas-Lara, Felipe García-Oliva, Salvador Sánchez-Carrillo, Javier Alcocer.

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
