## [Decision Letter · Decision Letter 0]

18 Apr 2024

PONE-D-23-38987Organic matter processing by potamoplankton in a large tropical river: relating elemental composition and potential carbon mineralizationPLOS ONE

Dear Dr. Alcocer, Thank you for submitting your manuscript to PLOS ONE. After careful consideration, we feel that it has merit but does not fully meet PLOS ONE’s publication criteria as it currently stands. Therefore, we invite you to submit a revised version of the manuscript that addresses the points raised during the review process. Please submit your revised manuscript by Jun 02 2024 11:59PM. If you will need more time than this to complete your revisions, please reply to this message or contact the journal office at plosone@plos.org. Please include the following items when submitting your revised manuscript:A rebuttal letter that responds to each point raised by the academic editor and reviewer(s). You should upload this letter as a separate file labeled 'Response to Reviewers'.A marked-up copy of your manuscript that highlights changes made to the original version. You should upload this as a separate file labeled 'Revised Manuscript with Track Changes'.An unmarked version of your revised paper without tracked changes. You should upload this as a separate file labeled 'Manuscript'.

We look forward to receiving your revised manuscript.

Kind regards,

Susmita Lahiri (Ganguly)

Academic Editor

PLOS ONE

Journal Requirements:

We require you to either present written permission from the copyright holder to publish these figures specifically under the CC BY 4.0 license, or remove the figures from your submission:

Reviewers' comments:

Reviewer's Responses to Questions

**Comments to the Author**

1. Is the manuscript technically sound, and do the data support the conclusions?

Reviewer #1: No

Reviewer #2: Yes

Reviewer #3: Partly

2. Has the statistical analysis been performed appropriately and rigorously? 

Reviewer #1: Yes

Reviewer #2: Yes

Reviewer #3: Yes

3. Have the authors made all data underlying the findings in their manuscript fully available?

Reviewer #1: Yes

Reviewer #2: Yes

Reviewer #3: Yes

4. Is the manuscript presented in an intelligible fashion and written in standard English?

Reviewer #1: Yes

Reviewer #2: Yes

Reviewer #3: Yes

5. Review Comments to the Author

**Reviewer #1:** The study by Cuevas-Lara et al. investigates the decomposition and mineralization of dissolved organic matter at three sites and two seasons in a tropical river and assesses the role of nutrients stoichiometry. The authors found very high losses of DOC with up to 90% after 6 days and high C-CO2 mineralization rates, especially in the lower regions of the river network and during the dry season. I think that the authors collected a valuable dataset; however, I would like to ask them to clarify the carbon mineralization method with the alkali trap before publication. Please find my comments to the different sections and specific comments with line numbers below.

Title: I am not sure that potamoplankton is the correct word that the authors use in the title and throughout the manuscript. The authors are studying the organic matter decomposition by primarily heterotrophic bacterioplankton. While potamoplankton refers to a river and its plankton, it also includes phyto- and zooplankton, which are not investigated in this study and are also not really discussed or mentioned. Therefore, I suggest that the authors try to use a different, more specific term here.

The abstract could be more concise to better reflect the study design and results. See my specific comments with line numbers below.

Methods and results: My main point, which I do not fully understand, concerns the measured mineralization rates with the alkali trap and the changes in the carbon pools. The C-CO2 rates: How did the authors treat the controls? As far as I can see, the distilled water controls showed a change of 8 mg l-1 day-1, which feels very high for distilled water. Where does the CO2 trapped in the alkali trap come from? I assume that the air can be excluded as this should be subtracted from the value with the blank value (which represents the CO2 in the air of the jar). There should be no microorganisms in distilled water that mineralize because firstly distilled water is often filtered and microbes do not survive in distilled water due to osmosis and secondly there is no carbon to mineralize. If the authors can say where the carbon comes from and if it is not microbes, then the value should also be subtracted from the other values, shouldn't it? I am asking because I wonder where all the CO2 comes from that was trapped in there. The C-CO2 rates range from 9.0 and 36.9 mg l−1 day−1. This means that a minimum of 54 mg C per liter in 6 days was mineralized (maximum would be at 221.4 mg per liter). How does it work when there was just a maximum of 22.6 mg l-1 of DOC (even when we consider also POC as a carbon source, which I definitely would), then we only have a maximum of around 23 mg l-1. So where does the carbon that is trapped there come from? Did the authors try to budget the changes in DOC/DIC/POC and the alkali trapped CO2? Does this all add up? I would ask the authors to clarify the method and check the measured values and/or calculations. Either I am misinterpreting something or something has gone wrong.

Discussion: I like the discussion that consists of three parts, in which the three hypotheses formulated in the introduction are discussed. I have not gone into all the details here yet because I want to see the answer to my main point about carbon budgeting and C-CO2 rates first. If something is wrong there, the whole story may change.

Specific comments:

L 35: Please add the time here for the incubation: “After 6-days incubation, …” or add at the end “…in six days/in 120 hours.” The incubation time is otherwise not mentioned in the abstract, but is important for the interpretation of DOC losses.

L 44: What exactly do the authors mean with “high C/N/P ratios in DOM in resource-rich environments”? Is there more C than nutrients and is the resource, the carbon or is the resource also nutrients? What exactly is a high C/N/P? High C/N and high C/P, this would mean C-enriched? I suggest that the authors clarify what exactly they mean by “resource-rich”.

L 44-47: The last sentence does not mention that the authors draw the conclusion about RIVER DOM processing. Although it is clear from before, I suggest that the authors consider adding river somewhere.

L 52-56: The high amounts of CO2 emitted from inland waters do not only come from microbial decomposition. A large proportion is actually CO2 that is respired in the soil and transported into streams and rivers. In addition, photochemical degradation also occurs, especially near the surface where about 10% of the carbon can be lost, as has been shown for boreal inland waters (Koehler et al. 2014). I suggest that the authors consider rewriting this first part to clarify that microbial decomposition is a process that may be important, perhaps even more so in tropical inland waters. However, there are other processes taking place that deserve mention here.

Koehler, B., T. Landelius, G. A. Weyhenmeyer, N. Machida, and L. J. Tranvik (2014), Sunlight-induced carbon dioxide emissions from inland waters, Global Biogeochem. Cycles, 28, 696–711, doi:10.1002/2014GB004850.

L 60-62: I am not sure I understand this sentence. The POM affects the reactions in the water? Do the authors mean the dissolved nutrients and respiration by the bacterioplankton or everything, including POM? I suggest firstly clarifying the sentence and secondly to put the last sentence of this paragraph first. Reference 7 was carried out on first-order streams, so what is said there also applies to headwaters where POM plays a greater role. I therefore suggest adding to the sentence that benthic processes play a greater role in small streams, and then ending the paragraph by saying that the water column in higher order streams is more important and deserves more attention.

L68/69: This sentence needs a reference.

L84/85: “which usually has higher requirements for N and P compared to that for C” What do the authors mean? Microbes generally need more N and P than C. I do not think that this is what they mean; because in general microbes still need more C than N or P, but the resources, in this case the DOM, often have a lower C/N or C/P ratio than necessary. Is that what the authors mean?

L 181-186: Samples were taken from three points in the river and then mixed into one 2L sterile plastic container or did the authors take triplicates here already? It is not clearly written if they took home one mixed sample or three samples. Please add a sentence with this information!

L192: “effectively incubated” What does this mean? I assume that the authors mean that the incubation starting after 5 days for 6 days corresponds to the usual time in which others carry out decomposition experiments. So these are still “natural” numbers. In general, I think it is not ideal to store the water samples for some time before incubation. However, I know that it is often not possible to perform these experiments in other ways due to transportation times and available personnel/equipment. Also, the authors have tried to keep this the same for all samples, which is the best they could do. Nevertheless, I believe that the authors should take this more into account in their discussion for the following reasons:

Firstly, the decomposition of organic matter by microbes is not linear. Rather, it is a very rapid decomposition of the more labile components at the beginning, which slows down over time as the less bioavailable components are degraded. The authors attempted to minimize this initial decomposition by storage at 4°C. However, this brings me to the second point. The change from 22-28°C in the river down to 4°C and then back to 25°C for incubation means that the community had to adapt to very different temperatures in a very short time. This should be mentioned in the text because it could affect the interpretation of the natural processes happening in the river. The third point the authors should think about is the oxygen conditions. They oxygenate their samples and therefore report aerobic decomposition, which is good. However, the sampling site in the lower basin (Centla) had very low oxygen concentrations during sampling (from 2.8 ± 1.1 - 5.8 ± 1.6 mg/L). A value below 3 mg/L is close to suboxic conditions where the microbial community may already be changing. At the very least, I would ask the authors to be aware of these changes when applying their results to what can/could happen in the river.

L213: extracted is usually a chemical treatment, what about “were removed from”?

L 376: The authors report the delta values for DOC in mg l-1 here. This is also the unit I am more familiar with when reporting DOC, however, authors decided to use µg l-1 in other places and, thus, it should be used consistently. I think this is a typo but please have a look that this is correctly reported.

L 423-427: Why did the authors do a stepwise regression model using the final concentrations at t120? How should they relate to the processing of carbon? I can imagine how the initial conditions and the changes could be related to the activity of the microbes. But I find the final concentrations more difficult. How should one interpret this?

**Reviewer #2: **General comments:

The manuscript title ‘’ Organic matter processing by potamoplankton in a large tropical river: relating elemental composition and potential carbon mineralization” investigates the interplay between dissolved organic carbon (DOM) and uptake of carbon (C), nitrogen (N), as well as carbon mineralization by freshwater planktonic communities in a significant river in southern part of Mexico. The authors sample three distinct locations across two seasons along the river. Capturing the effects of varied vegetation types and a unique seasonal regime typical of the wet and dry season, highlighting the influence of a river regime and DOM on microbial metabolic activity.

Overall, the authors have proficiently described the methodologies employed and chosen an appropriate data treatment strategy to address the research questions effectively. The presentation of the results and figures aligns well the manuscript objectives providing a clear and coherent narrative.

However, there are areas of concerns regarding the methodology that required further elucidation, for example my main concern is how representative is the sampling of the seasonal conditions of the river at the three localities. Additionally, some minor edits are necessary to enhance the articles readability and address verbosity issues identified by the reviewer. Particularly within the introduction and discussion sections. The article leaves a general sensation of unevenness, there are some very well written and articulated paragraphs that usually follow a not so well written one. Despite these issues, the manuscript is recommended for publication, contingent on the resolution of the issues specified in the following specific comments section.

Specific comments:

Abstract (Suggested changes)

Line 32 to 34: Difficult to follow section, recommended correction: Our study investigated changes in the composition and concentration of dissolved organic matter (DOM) and evaluated carbon dioxide (CO2) production rates through laboratory experiments. We compared three sites representing the middle and lower basins, including their transitional zones, during both the rainy and dry seasons.

Line 43 to 44: Our findings indicate that microbial metabolism operates with reduced efficiency in environments DOM rich environments, particularly when faced with high carbon/nitrogen/phosphorus (C/N/P) ratios.

Introduction:

Line 50 to 54: Recent studies indicate that inland waters contribute approximately 0.9 PgC yr-1 to the oceans globally. This figure is significantly lower than the 2.9 PgC yr-1 that originates from terrestrial sources and the 1.9 PgC yr-1that is released into the atmosphere. These differences suggest that the decomposition of organic matter by heterotrophic prokaryotes plays a crucial role, as they mineralize organic matter into CO2 through respiration, thereby affecting the evasion of carbon (C) from rivers.

Line 57 to 59: The influence of dissolved organic matter (DOM) and inorganic nutrients on carbon (C) mineralization is diverse. Inputs of nitrogen (N) and phosphorus (P) can either elevate respiration rates, reduce biomass, or elicit negligible reactions among planktonic and benthic microbial communities.

Line 62 to 66: Recent findings indicate that in large rivers (fifth order or above), these metabolic processes predominantly take place within the water column rather than the riverbed. This is attributed to the enhanced surface area for contact between the water and suspended sediments found further downstream. Conversely, in smaller upland streams, the assimilation of nutrients from sediment pores amplifies nutrient cycling within the riverbed compared to the water column.

Line 78 to 80: Good ending for the paragraph.

Line 85 to 90: For instance, bacteria prioritize enzyme production over somatic growth to access scarce nutrients. The carbon/nitrogen/phosphorus (C/N/P) ratios in organic matter significantly affect microbial decomposition rates and carbon mineralization, with these ratios varying according to the source of organic matter along rivers. Autochthonous inputs, like phytoplankton, exhibit low C/N and C/P ratios, whereas allochthonous inputs, such as wood and leaves, feature high C/N and C/P ratios.

Line 103: Change to present tense, word “aimed” to “aims:” In methods is ok to use past.

Line 106: Same and line 103.

Line 110: I suggest to change “We anticipated “to “We hypothesized”.

Line 116 to 118: By identifying the factors influencing microbial C mineralization along the Usumacinta River, we contribute to a better understanding of the carbon dynamics involved in dissolved organic matter (DOM) processing within a tropical riverine ecosystem.

Materials and methods:

Main question to address is how can I be sure that the sampling represents accurately the seasonal conditions of the river, given that to my understanding there was only one sampling done per location with no replicates? Its need to be urgently discussed further the likelihood of the values represented on this study not being just a combination of the factors for that particular day. My last question is in regard to the depth of sampling. Why sampling only in surface waters, would it be conceivable to expect increase metabolic activity of the planktonic community closer to the bottom, especially during the dry season. Or rivers in general are so well mix that this is not an issue? Perhaps something to consider in the discussion as well.

Line 135 to 137: What happened with the dry season description? Very nice description of the site and the rainy season. But the description of the dry season needs development. Other readers might not be aware of its conditions and I believe there could be other interesting facts to highlight besides when it happens. Also, in line 137 the word inundations have more than one meaning probably the word floodings is more appropriate.

Line 209: Replace “discount” with “discard”.

Line 295 to 313: The PCA employed Box-Cox transformations (with λ = 0.75) to normalize the data, along with a Euclidean similarity index and a data correlation matrix to ensure normality, linearity, and standardization in the ordination process. The limit of detection (LOD) for N and P, set at 0.1 and 0.3 mg/L respectively, was applied within the PCA for values falling below these thresholds.

A subsequent PCA focused on the dissolved fraction differences, utilizing a correlation matrix and an iterative imputation method to account for missing data. This analysis linked the components identified in the first PCA and the carbon dioxide (C-CO2) production rate with those derived from the second PCA. For both PCAs, significant components were selected based on expected eigenvalues using a random model approach (Broken Stick method).

We incorporated a linear mixed-effects model to evaluate the influence of C/N/P ratios in the DOM on the C-CO2 production rate. This model treated seasons, sites, and C/N/P ratios as fixed effects while considering the sample bottle as a random effect to adjust for sample interdependence.

Results

This section needs some re-writing and careful consideration on how to improve the flow of how the results are presented. It would be lengthy to do specific suggestions. However, here I briefly suggest how the flow could be improved:

Line 326 to 332: Dry season: The concentrations of POC and TSS constituted approximately 10 ± 1% (mean ± standard deviation) of the %OC. In the Lacantún and Balancán sites (representing the middle and transitional areas), the DOC concentration was roughly 21,000 µg l-1, nearly double that of the Centla site (lower area), which registered 12,620 µg l-1. Notably, the concentration of NO3- decreased progressively downstream, dropping below the LOD in Centla, where the concentration of NH4+ reached its peak. The SRP concentration was below the LOD. Downstream, the DOC/DON and DOC/DOP ratios declined, a trend attributed to the rising concentrations of DON and DOP coupled with a decrease in DOC levels.

Discussion:

Line 442 to 445: Similar studies have also identified elevated respiration rates in organic substrates with a surplus of C and a scarcity of P and N, through microbial experiments that involve culturing and incubating heterotrophic aquatic bacteria. These increased rates are expected to result in heightened C mineralization, a finding that aligns with the outcomes observed in our study.

Line 448: I believe the abbreviation CUE has not been used before.

Line 457: Very interesting point, perhaps this could be expanded a bit more and check if other studies have found the same.

**Reviewer #3: **I think this paper has demonstrated a well and detail work for the carbon and nitrogen change in POM intersecting the different conditions of the upstream and downstream flows and the seasons ( dry and rainy). And the incubation was used to detect the function of microbial mineralization. This is a good combination to the biological and physical/ meteorology, but I think the research difficulty is not too high from the perspective of method. And the conclusion is not too breakthrough discoverie or impressive view.

In addition, the organic matter processing by potamoplankton can not be found in the research except the filter process with 0.7μm pore diameter of filter membrane to filter the particulate fraction. Maybe I can not grasp your method because my academic level is limited, so i think the connection between the potamoplankton and the organic matter processing should be be analyzed in the paper.

6. PLOS authors have the option to publish the peer review history of their article (what does this mean?). If published, this will include your full peer review and any attached files.

Reviewer #1: No

Reviewer #2: **Yes: **Paulo F. Lagos

Reviewer #3: No

---

## [Author Response · Author response to Decision Letter 0]

23 Aug 2024

Review on manuscript PONE-D-23-38987

Authors' responses to the journal and reviewers’ comments:

Journal Requirements:

Authors' answer: We have attended the requirements of the PLOSONE style.

Authors' answer: We included the following sentence in the methods section: No permits are required for scientific fieldwork in non-natural protected areas (e.g., Usumacinta River). 

Authors' answer: The code used in the manuscript was not author-generated code nor central to the manuscript.

Authors' answer: Figure 1 was modified with a base map obtained from web services with data and files from USGS with the public domain as suggested by the PLOS ONE journal (please see https://qms.nextgis.com/geoservices/2132/). We have added the legend “Map services and data available from U.S. Geological Survey, National Geospatial Program.” to Figure 1 to fulfill the terms of Use/Licensing of map services asked by the USGS. 

Reviewer #1

Reviewer #1: The study by Cuevas-Lara et al. investigates the decomposition and mineralization of dissolved organic matter at three sites and two seasons in a tropical river and assesses the role of nutrients stoichiometry. The authors found very high losses of DOC with up to 90% after 6 days and high C-CO2 mineralization rates, especially in the lower regions of the river network and during the dry season. I think that the authors collected a valuable dataset; however, I would like to ask them to clarify the carbon mineralization method with the alkali trap before publication.

Authors' general answer: We appreciate the reviewer’s comments, which have greatly improved our manuscript. The comments have been addressed throughout the manuscript. Please see the answer to each comment.

Title: I am not sure that potamoplankton is the correct word that the authors use in the title and throughout the manuscript. The authors are studying the organic matter decomposition by primarily heterotrophic bacterioplankton. While potamoplankton refers to a river and its plankton, it also includes phyto- and zooplankton, which are not investigated in this study and are also not really discussed or mentioned. Therefore, I suggest that the authors try to use a different, more specific term here.

Authors' answer: We modified the title according to the referee’s comment to:

“Organic matter processing by heterotrophic bacterioplankton in a large tropical river: relating elemental composition and potential carbon mineralization”

The term heterotrophic bacterioplankton is now uniformly used throughout the document.

The abstract could be more concise to better reflect the study design and results. See my specific comments with line numbers below.

Authors' answer: We have adjusted the abstract to agree with the referee’s comments.

Methods and results: My main point, which I do not fully understand, concerns the measured mineralization rates with the alkali trap and the changes in the carbon pools. The C-CO2 rates: How did the authors treat the controls? As far as I can see, the distilled water controls showed a change of 8 mg l-1 day-1, which feels very high for distilled water. Where does the CO2 trapped in the alkali trap come from? I assume that the air can be excluded as this should be subtracted from the value with the blank value (which represents the CO2 in the air of the jar). There should be no microorganisms in distilled water that mineralize because firstly distilled water is often filtered, and microbes do not survive in distilled water due to osmosis and secondly there is no carbon to mineralize. If the authors can say where the carbon comes from and if it is not microbes, then the value should also be subtracted from the other values, shouldn't it? I am asking because I wonder where all the CO2 comes from that was trapped in there. The C-CO2 rates range from 9.0 and 36.9 mg l−1 day−1. This means that a minimum of 54 mg C per liter in 6 days was mineralized (maximum would be at 221.4 mg per liter). How does it work when there was just a maximum of 22.6 mg l-1 of DOC (even when we consider also POC as a carbon source, which I definitely would), then we only have a maximum of around 23 mg l-1. So where does the carbon that is trapped there come from? Did the authors try to budget the changes in DOC/DIC/POC and the alkali trapped CO2? Does this all add up? I would ask the authors to clarify the method and check the measured values and/or calculations. Either I am misinterpreting something or something has gone wrong.

Authors' answer: As the referee mentions, exposing distilled water to air before incubation could have introduced CO2 into the control samples, which was later transferred to the alkali traps. Accordingly, as the reviewer suggested, we have subtracted the captured CO2 in controls from the C-CO2 emission rates in the samples. We have clarified this procedure in the material and methods section (in the “Potential carbon mineralization” section). We apologize for these oversights and thank the reviewer for their observation. The new sentence in that section states:

“The C–CO2 fluxes average in controls (10.2 ± 2.5 mg l−1 day−1 in the dry season and 8.4 ± 3.4 mg l−1 day−1 in the rainy season) were subtracted from those for samples due to the atmospheric CO2 that could enter the water before the incubations considering the solubility of CO2 [57] and the water acidification in distilled water [58]”

Discussion: I like the discussion that consists of three parts, in which the three hypotheses formulated in the introduction are discussed. I have not gone into all the details here yet because I want to see the answer to my main point about carbon budgeting and C-CO2 rates first. If something is wrong there, the whole story may change.

Authors' answer: We appreciate the reviewer’s comment. Based on the minor changes in the magnitude of the C–CO2 rates with the adjustment mentioned before, we made minor changes to the discussion, as the patterns remained similar.

Specific comments:

L 35: Please add the time here for the incubation: “After 6-days incubation, …” or add at the end “…in six days/in 120 hours.” The incubation time is otherwise not mentioned in the abstract, but is important for the interpretation of DOC losses.

Authors' answer: The information has been added in the first sentence in parentheses:

“After incubation (120 h at 25 °C), the DOM had lost between 25 % and 89 % of its C.”

L 44: What exactly do the authors mean with “high C/N/P ratios in DOM in resource-rich environments”? Is there more C than nutrients and is the resource, the carbon or is the resource also nutrients? What exactly is a high C/N/P? High C/N and high C/P, this would mean C-enriched? I suggest that the authors clarify what exactly they mean by “resource-rich”.

Authors' answer: We agree. We clarified the information in the re-written sentence:

“Our findings indicate that microbial metabolism operates with reduced efficiency in C -rich environments like forests, particularly when faced with high C/N and C/P ratios in DOM”

L 44-47: The last sentence does not mention that the authors draw the conclusion about RIVER DOM processing. Although it is clear from before, I suggest that the authors consider adding river somewhere.

Authors' answer: We have added “river DOM processing” at the end of the sentence.

L 52-56: The high amounts of CO2 emitted from inland waters do not only come from microbial decomposition. A large proportion is actually CO2 that is respired in the soil and transported into streams and rivers. In addition, photochemical degradation also occurs, especially near the surface where about 10% of the carbon can be lost, as has been shown for boreal inland waters (Koehler et al. 2014). I suggest that the authors consider rewriting this first part to clarify that microbial decomposition is a process that may be important, perhaps even more so in tropical inland waters. However, there are other processes taking place that deserve mention here.

Koehler, B., T. Landelius, G. A. Weyhenmeyer, N. Machida, and L. J. Tranvik (2014), Sunlight-induced carbon dioxide emissions from inland waters, Global Biogeochem. Cycles, 28, 696–711, doi:10.1002/2014GB004850.

Authors' answer: As far as we know, the differences in C between terrestrial loads and ocean exports were attributed (e.g., Maranger et al. 2018) to respiratory C losses through differences in the C:N:P stoichiometry. Therefore, we highlighted organic matter processing (i.e., decomposition) by heterotrophic prokaryotes in the C evasion. Other authors have also supported this conclusion (e.g., Mineau et al., 2016). However, we agree that other relevant processes take place and should be mentioned, so we have added a clarification to the new sentence:

“These differences suggest that the decomposition of organic matter by heterotrophic prokaryotes, along with other abiotic processes like photomineralization and terrestrial inputs of CO2 [2], plays a crucial role in the fluvial C evasion, as they mineralize organic matter into CO2 through respiration, especially in the tropics [3,4]”

L 60-62: I am not sure I understand this sentence. The POM affects the reactions in the water? Do the authors mean the dissolved nutrients and respiration by the bacterioplankton or everything, including POM? I suggest firstly clarifying the sentence and secondly to put the last sentence of this paragraph first. Reference 7 was carried out on first-order streams, so what is said there also applies to headwaters where POM plays a greater role. I therefore suggest adding to the sentence that benthic processes play a greater role in small streams, and then ending the paragraph by saying that the water column in higher order streams is more important and deserves more attention.

Authors' answer: We have clarified that POM affects the availability of dissolved nutrients for bacterioplankton, which influences microbial respiration. We have also added another reference stating this idea (Stelzer et al., 2003). We have checked the text and addressed the order of ideas, and the new paragraph states:

“The influence of dissolved organic matter (DOM) and inorganic nutrients on C mineralization should change along the rivers. Inputs of nitrogen (N) and phosphorus (P) can either elevate respiration rates, reduce biomass, or elicit negligible reactions among planktonic and benthic microbial communities [6–8]. In small upland streams, the assimilation of nutrients from sediment pores amplifies nutrient cycling within the riverbed compared to the water column [9]. Furthermore, the quantity of particulate organic matter (POM) affects the influence of dissolved nutrients on microbial respiration because microbial extracellular enzymes produce monomers with the available nutrients by degrading large biomolecules in the POM [10,11]. However, recent findings indicate that these metabolic processes predominantly occur within the water column rather than the riverbed in large rivers (fifth order or above). This is attributed to the enhanced surface area for contact between the water and suspended sediments found further downstream [12,13]. Thus, the influence of stream order on microbial processing of organic matter in the water column needs more attention from researchers.” 

L68/69: This sentence needs a reference.

Authors' answer: The reference was added (i.e., Battin et al., 2008) as follows:

“The availability of organic matter changes with landscape characteristics along basins and according to precipitation levels [5]”

L84/85: “which usually has higher requirements for N and P compared to that for C” What do the authors mean? Microbes generally need more N and P than C. I do not think that this is what they mean; because in general microbes still need more C than N or P, but the resources, in this case the DOM, often have a lower C/N or C/P ratio than necessary. Is that what the authors mean?

Authors' answer: We agree with the referee. We changed the word “requirement” to “limited” and gave context on resource stoichiometry. We have modified the texts as follows:

“Microbes must invest energy to adjust these elemental ratios to maintain homeostasis within their biomass, whose growth is usually limited by N, P, or both compared to C according to resource stoichiometry [24–26].”

L 181-186: Samples were taken from three points in the river and then mixed into one 2L sterile plastic container or did the authors take triplicates here already? It is not clearly written if they took home one mixed sample or three samples. Please add a sentence with this information!

Authors' answer: We used a composed sample of the three points, so we referred to the first option (with mixing). Thus, we have changed “The samples were used to fill 2…” to “The samples were mixed and used to fill…” for clarity.

L192: “effectively incubated” What does this mean? I assume that the authors mean that the incubation starting after 5 days for 6 days corresponds to the usual time in which others carry out decomposition experiments. So these are still “natural” numbers. In general, I think it is not ideal to store the water samples for some time before incubation. However, I know that it is often no

---

## [Decision Letter · Decision Letter 1]

24 Sep 2024

Organic matter processing by heterotrophic bacterioplankton in a large tropical river: relating elemental composition and potential carbon mineralization

PONE-D-23-38987R1

Dear Dr. Alcocer,

We’re pleased to inform you that your manuscript has been judged scientifically suitable for publication and will be formally accepted for publication once it meets all outstanding technical requirements.

Kind regards,

Susmita Lahiri (Ganguly)

Academic Editor

PLOS ONE

Additional Editor Comments (optional):

Reviewers' comments:

Reviewer's Responses to Questions

**Comments to the Author**

1. If the authors have adequately addressed your comments raised in a previous round of review and you feel that this manuscript is now acceptable for publication, you may indicate that here to bypass the “Comments to the Author” section, enter your conflict of interest statement in the “Confidential to Editor” section, and submit your "Accept" recommendation.

Reviewer #4: All comments have been addressed

Reviewer #5: All comments have been addressed

2. Is the manuscript technically sound, and do the data support the conclusions?

Reviewer #4: Partly

Reviewer #5: Yes

3. Has the statistical analysis been performed appropriately and rigorously? 

Reviewer #4: N/A

Reviewer #5: Yes

4. Have the authors made all data underlying the findings in their manuscript fully available?

Reviewer #4: Yes

Reviewer #5: Yes

5. Is the manuscript presented in an intelligible fashion and written in standard English?

Reviewer #4: No

Reviewer #5: Yes

6. Review Comments to the Author

Reviewer #4: English of manuscript need to be improved. Authors have already addressed the queries raised by the reviewer. Though, it seems that methods section should be improved more by adding few more techniques.

Reviewer #5: (No Response)

7. PLOS authors have the option to publish the peer review history of their article (what does this mean?). If published, this will include your full peer review and any attached files.

Reviewer #4: No

Reviewer #5: **Yes: **Dr. Muthuraju R.

---

## [Editor Report · Acceptance letter]

31 Oct 2024

PONE-D-23-38987R1 

PLOS ONE

Dear Dr. Alcocer, 

I'm pleased to inform you that your manuscript has been deemed suitable for publication in PLOS ONE. Congratulations! Your manuscript is now being handed over to our production team.

Kind regards, 

on behalf of

Dr. Susmita Lahiri (Ganguly) 

Academic Editor

PLOS ONE